# Association of Permanent Vascular Access Dysfunction with Subsequent Risk of Cardiovascular Disease: A Population-Based Cohort Study

**DOI:** 10.3390/jpm12040598

**Published:** 2022-04-08

**Authors:** Tung-Wei Hung, Sheng-Wen Wu, Jeng-Yuan Chiou, Yu-Hsun Wang, Yu-Chan Liao, Cheng-Chung Wei

**Affiliations:** 1School of Medicine, Chung Shan Medical University, Taichung 402306, Taiwan; a6152000@ms34.hinet.net (T.-W.H.); s41111.tw@yahoo.com.tw (S.-W.W.); 2Institute of Medicine, Chung Shan Medical University, Taichung 402306, Taiwan; 3Division of Nephrology, Department of Internal Medicine, Chung Shan Medical University Hospital, Taichung 40201, Taiwan; n8705039@gmail.com; 4School of Health Policy and Management, Institute of Medicine, Chung Shan Medical University, No. 110, Sec. 1, Jianguo N Road, South District, Taichung 40201, Taiwan; drchiou@hotmail.com; 5Department of Medical Research, Chung Shan Medical University Hospital, Taichung 40201, Taiwan; cshe731@csh.org.tw; 6Division of Allergy, Immunology, and Rheumatology, Department of Medicine, Chung Shan Medical University Hospital, Taichung 40201, Taiwan

**Keywords:** permanent vascular access, arteriovenous fistula, arteriovenous grafts, dialysis, cardiovascular disease

## Abstract

A functional permanent vascular access (VA) is required to perform a successful hemodialysis procedure. Hemodialysis VA dysfunction is a major cause of morbidity and hospitalization in the hemodialysis population. Cardiovascular disease (CVD) is the leading cause of death in patients receiving chronic hemodialysis. Information about CVD associated with hemodialysis VA dysfunction is unclear. We analyzed the association between dialysis VA dysfunction and the risk of developing CVD in hemodialysis patients. This nationwide population-based cohort study was conducted using data from the National Health Insurance Research Database in Taiwan. One million subjects were sampled from 23 million beneficiaries and data was collected from 2000 to 2013. Patients with end-stage renal disease who had received permanent VA construction and hemodialysis and were aged at least 20 years old from 2000 to 2007 were included in the study population. The primary outcome was CVD, as defined by ICD-9-CM codes 410–414 and 430–437. A total of 197 individuals with permanent VA dysfunction were selected as the test group, and 100 individuals with non-permanent VA dysfunction were selected as the control group. Compared with the control group, the adjusted hazard ratio of CVD for the VA dysfunction group was 3.05 (95% CI: 1.14–8.20). A Kaplan–Meier analysis revealed that the cumulative incidence of CVD was higher in the permanent VA dysfunction group than in the comparison group. Permanent VA dysfunction is significantly associated with an increased risk of subsequent CVD.

## 1. Introduction

Options for hemodialysis access include catheters, arteriovenous grafts (AVGs), and arteriovenous fistulas (AVFs). An AVF is the most common and preferred method of vascular access (VA) for chronic hemodialysis patients due to high blood flow rate, patency, and low infection risk. Permanent VAs include both native AVFs and AVGs. A functional permanent VA is required to perform a successful hemodialysis procedure. Venous stenosis leading to thrombosis in the peri-anastomotic region of an AVF or at the graft-vein anastomosis of a polytetrafluoroethylene (PTFE) graft is the primary cause of VA dysfunction [1]. Many studies have excluded primary failures in analyses of patency, while other studies have reported patency only at specific time points (i.e., six months or one year) [2,3,4]. A large, randomized controlled trial, published by the National Institutes of Health Dialysis Access Consortium in 2008, reported that 60% of AVFs failed to sufficiently mature for successful dialysis four to five months after creation [5]. Hemodialysis VA dysfunction is a major cause of morbidity and hospitalization in the hemodialysis population [6,7]. Permanent VA dysfunction can also be caused by injury-causing factors other than vascular stenosis. For example, hemodynamic shear stress and cannulation injury during hemodialysis, vascular manipulation (balloon angioplasty or surgery) [7,8,9], and inflammation and oxidative stress associated with uremia can induce neointimal hyperplasia and vascular injury [10,11]. These novel, nontraditional risk factors are also common underlying causes of VA dysfunction in hemodialysis patients [1,8,12].

Cardiovascular disease (CVD) is the leading cause of death in patients receiving chronic hemodialysis [13,14,15]. Not only are several traditional risk factors present in CVD, but novel, nontraditional risk factors associated with CVD are also emerging in patients with end-stage renal disease (ESRD). Cardiovascular changes that are secondary to renal dysfunction—including fluid overload, uremic cardiomyopathy, chronic kidney disease-mineral-bone-disorder, anemia, altered lipid metabolism, accumulation of gut microbiota-derived uremic toxins (e.g., trimethylamine *N*-oxidase), inflammation, oxidative stress, and other aspects of the *uremic milieu*—contribute to a high risk of CVD in the dialysis population [16,17].

CVD is very common in hemodialysis patients and poses a high risk of mortality. The artificially created structures of permanent VA in hemodialysis resemble the small-to-medium vessels in the cardiovascular system. VA dysfunction has been postulated to be associated with CVD in hemodialysis patients [18,19]. Due to a lack of research on the epidemiologic relationship between permanent VA dysfunction and CVD in end-stage renal disease, the present nationwide longitudinal cohort study was conducted to explore whether patients with permanent arteriovenous access dysfunction are prone to the subsequent development of CVD.

## 2. Materials and Methods

### 2.1. Data Source

The present study was a retrospective cohort study using the National Health Insurance Research Database (NHIRD), which enrolled almost 99% of the population of 23 million beneficiaries in Taiwan. This database included all insurance claims data, including data concerning outpatient visits, emergencies, and hospitalization. One million subjects were sampled from the 23 million beneficiaries, and data was collected from 2000 to 2013. The sampled database was de-identified, and the study was approved by the Institutional Review Board of Chung Shan Medical University Hospital.

### 2.2. Study Group and Outcome Measurement

In the NHIRD, diseases are classified according to the International Classification of Diseases, 9th Revision, Clinical Modification (ICD-9-CM). Patients with ESRD (ICD-9-CM: 585) who had received arteriovenous VA construction (order code: 69032B, 69032C, 69034C) and hemodialysis (order code: 58001C, 58019C-58029C) and were aged at least 20 years old from 2000 to 2007 were included in the study population (Figure 1). Definitions and reporting of patency have differed significantly in arteriovenous access studies. In order to ensure that observed CVD was newly onset, we excluded patients who had undergone arteriovenous VA before the hemodialysis date. Further, we excluded patients who had experienced dysfunction of arteriovenous VA with stenosis or thrombosis within half a year after the first hemodialysis date. Permanent VA dysfunction was defined as stenosis or thrombosis of arteriovenous VA needing radiological or surgical intervention to facilitate or maintain patency more than 0.5 to 5 years after the hemodialysis date. The non-dysfunction group had never needed radiological or surgical intervention to facilitate or maintain patency of arteriovenous VA past half a year after the hemodialysis date. The index date was the first date five years after hemodialysis treatment.

The outcome variable was defined as a diagnosis of CVD (ICD-9-CM: 410–414, 430–437) accompanied by at least two outpatient visits or one hospitalization. The study was followed up until the occurrence of CVD, the date 31 December 2013, or withdrawal from the national insurance system, whichever occurred first. To confirm newly onset CVD, we excluded those who had been diagnosed with CVD (ICD-9-CM: 410–414, 430–438) before the index date.

### 2.3. Covariates

The baseline characteristics were age, gender, hypertension (ICD-9-CM: 401–405), hyperlipidemia (ICD-9-CM: 272.0–272.4), diabetes (ICD-9-CM: 250), chronic obstructive pulmonary disease (ICD-9-CM: 491, 492, 496), autoimmune disease (ICD-9-CM: 710.0, 714.0, 720.0), asthma (ICD-9-CM: 493), chronic liver disease (ICD-9-CM: 571), and hyperparathyroidism (ICD-9-CM: 588.8). These comorbidities were defined before the index date, within five years, and accompanied by at least two outpatient visits or one hospitalization. In addition, warfarin, corticosteroids, statin, and aspirin were included if used before the index date, within five years, and for at least 30 days.

### 2.4. Statistical Analysis

Comparison between the permanent VA dysfunction group and the non-dysfunction group was conducted using a chi-square test, Fisher’s exact test, or Student’s *t* test, as deemed appropriate. A Kaplan–Meier analysis was used to calculate the cumulative incidence of CVD, and a log-rank test was used to test the statistical significance. The Cox proportional hazard model was used to estimate the hazard ratio of CVD between the dysfunction group and the non-dysfunction group and was adjusted for age, gender, comorbidities, and medications. SPSS Statistics software version 18.0 (SPSS Inc., Chicago, IL, USA) was used for statistical analysis, and *p* values less than 0.05 were defined as significant.

## 3. Results

### 3.1. Characteristics of Study Patients

A total of 197 individuals with permanent VA dysfunction and 100 individuals with non-permanent VA dysfunction were selected based on our inclusion and exclusion criteria (Figure 1). Table 1 shows comparisons of the demographic data of the permanent VA dysfunction group with that of the non-permanent VA dysfunction group. There were more female patients than male patients, and most of the patients were aged 40–65 years. The permanent VA dysfunction group exhibited higher risks of comorbidities with hyperlipidemia and heart failure as well as a higher ratio of warfarin and statin administration. The mean follow-up times in the permanent VA dysfunction and non-permanent VA dysfunction groups were 5.4 and 5.1 years, respectively.

### 3.2. Risk of Cardiovascular Disease

Among the 36 patients with CVD events, 26 patients had coronary artery disease (ICD-9-CM = 410–414) and 10 patients had stroke (ICD-9-CM = 430–437). The incidence rates of CVD in the permanent VA dysfunction and comparison groups were 29.2 and 9.8 per 1000 person years, respectively. Compared with the control group, the adjusted hazard ratio for CVD for the permanent VA dysfunction group was 3.05 (95% CI: 1.14–8.20; Table 2). The Kaplan–Meier analysis (Figure 2) revealed that the cumulative incidence of CVD was higher in the permanent VA dysfunction group than in the comparison group (*p* = 0.020). The age-subgroup analysis showed that the permanent VA dysfunction group had an increased risk of CVD in the 20–65 age group (HR = 4.37, 95% CI: 1.03–18.51).

### 3.3. Risk of Cardiovascular Disease among Patients with and without Vascular Access Occlusion and Subgroup Specific Characteristics

We performed subgroup analysis between the vascular access occlusion group and the non-occlusion group. An increased risk of CVD was statistically observed in the occlusion group for patients aged 20–65 years (4.37-fold). An increased risk of CVD was also observed in patients without warfarin (2.74-fold) and statin (3.54-fold) administration in the occlusion group (Table 3).

## 4. Discussion

Although permanent VA has been recommended as the preferred VA method, VA dysfunction remains a frequent cause of morbidity in patients receiving chronic hemodialysis [20,21]. In addition to cardiovascular mortality, cardiovascular morbidities are commonly observed in these dialysis patients [18]. Investigation of the relationship between arteriovenous access dysfunction and CVD will have direct implications for patient care.

In this nationwide population-based cohort study, we analyzed the outcomes of 297 patients with a functional permanent VA that had successfully matured and had been used for dialysis for six months after first use. In comparison with the control group, patients with permanent VA dysfunction had a 3.05-fold higher incidence of CVD. Furthermore, the association of permanent VA dysfunction with CVD was found to be most significant in patients aged 20–65 years. It is generally accepted that CVD is the leading cause of death across all age groups of patients with ESRD [22]. It will potentially allow us to identify patients at higher risk of CVD and consider surveillance of functional VA.

The two main causes of permanent VA dysfunction are initial failure to mature and, later, venous stenosis [7]. Because the main objective of the present study was to investigate permanent VA dysfunction after maturation and explore its association with CVD, patients that required intervention (i.e., surgical or percutaneous endovascular intervention) to achieve maturation within six months were specifically excluded. Venous stenosis at the VA or graft vein anastomosis leading to thrombosis is the primary cause of VA failure. Venous stenosis is primarily due to venous neointimal hyperplasia in both AVGs and AVFs. The pathogenesis of venous neointimal hyperplasia includes oxidative stress, endothelial dysfunction, uremia, and inflammation [8]. The predisposing factors associated with VA dysfunction include female sex, old age, smoking, greater body mass index, diabetes, hyperlipidemia, coronary artery disease, peripheral vascular disease, surgical skill levels, small vessel or anatomical anomalies, and a hypercoagulable state [23,24,25,26,27]. CVD risk factors among hemodialysis patients can be divided into traditional risk factors (e.g., age, male sex, hypertension, diabetes, dyslipidaemia, and physical inactivity) and uremia-related risk factors (e.g., anemia, hyperhomocysteinaemia, chronic kidney disease-mineral-bone-disorder, oxidative stress, malnutrition, and chronic inflammation) [17]. Uremic toxins impact the functions of cells involved in myocardial and vessel functions, such as leucocytes, endothelial cells (ECs), smooth muscle cells, and platelets [28]. The pathogenesis of hemodialysis VA dysfunction was similar to that of CVD. It is possible that there is an association between VA dysfunction and CVD. The relationship between VA and CVD exists at different levels, ranging from inflammation promoting atherosclerotic disease to vascular remodelling changes of stenosis formation and left ventricular hypertrophy [29]. The authors suggest that CVD surveillance in patients with permanent VA dysfunction is important.

Patients with AVF and AVG complications often require temporary placement of a central venous catheter, which is the least desirable type of VA due to significantly higher rates of catheter-associated bacteremia, fatal infections, and cardiovascular events; inadequate solute clearance; and higher all-cause mortality [30,31,32]. The inhibition of leucocyte activity in hemodialysis patients may compromise infection response and trigger micro-inflammation and, consequently, atherosclerosis. Moreover, uremic toxins induce enhancement of leucocyte oxidative activity, upregulation in leucocyte–endothelial interactions, and infiltration of macrophages and monocytes into vascular atherosclerotic lesions [33], all of which may be related to subsequent CVD.

The systematic review of random control trials investigating the effects of aspirin and warfarin therapy on VA outcomes did not find any beneficial effect to increase the patency of AVFs or AVGs [34,35]. Several retrospective analyses suggested no significant benefits of statin use with respect to the primary or secondary patency of AVFs or AVGs [36], stenosis formation [37] or cumulative access survival (after excluding primary failure) [36,38]. In the present study, a prominent risk of CVD was observed in the patients with hyperlipidemia (adjusted HR: 3.33) as well as those administered aspirin (adjusted HR: 2.94) in comparison to the non-arteriovenous access dysfunction population.

Jeong examined long-term VA patency in an Asian hemodialysis patient population with confirmed first VA placement, stratified by age (<65 years vs. ≥65 years), and showed that increased age was associated with shorter primary patency [39]. In CKD patients on hemodialysis, older age confers a CVD risk that parallels the relationship described in the general population [40]. In the study, the association between VA dysfunction and CVD was valid even after adjustment of age parameters. The present results suggested that patients within 20–65 years of age require increased surveillance of their VA patency, as permanent VA dysfunction with CVD was found to be most significant in this age group.

The present study had some limitations. First, this study was retrospective in nature and therefore did not offer a randomized comparison of autogenous fistulas and prosthetic grafts. AVFs are considered superior to AVGs because of their longer secondary patency after successful cannulation for dialysis. We could not account for factors that might have affected patency, such as conduit quality, medication use, biologic or synthetic graft subtypes, the precise cause of access failure or death, or the surgeon’s experience and skill. Other important factors not included in our data sources (e.g., uremic signs and symptoms at the time of hemodialysis initiation and vessel diameter and quality) could have impacted the results; these factors might have accounted for the differences in our outcomes. Prospective studies and clinical trials are needed to further clarify the association that can be performed in public health strategies to predict CVD in hemodialysis patients.

In conclusion, we found that permanent VA dysfunction is significantly associated with an increased risk of subsequent CVD among patients on maintance hemodialysis under 65 years old. In the high-risk cardiovascular profile of hemodialysis patients, monitoring and surveillance of permanent VA should be employed in order to identify a dialysis access dysfunction for association with subsequent CVD.

## Figures and Tables

**Figure 1 jpm-12-00598-f001:**
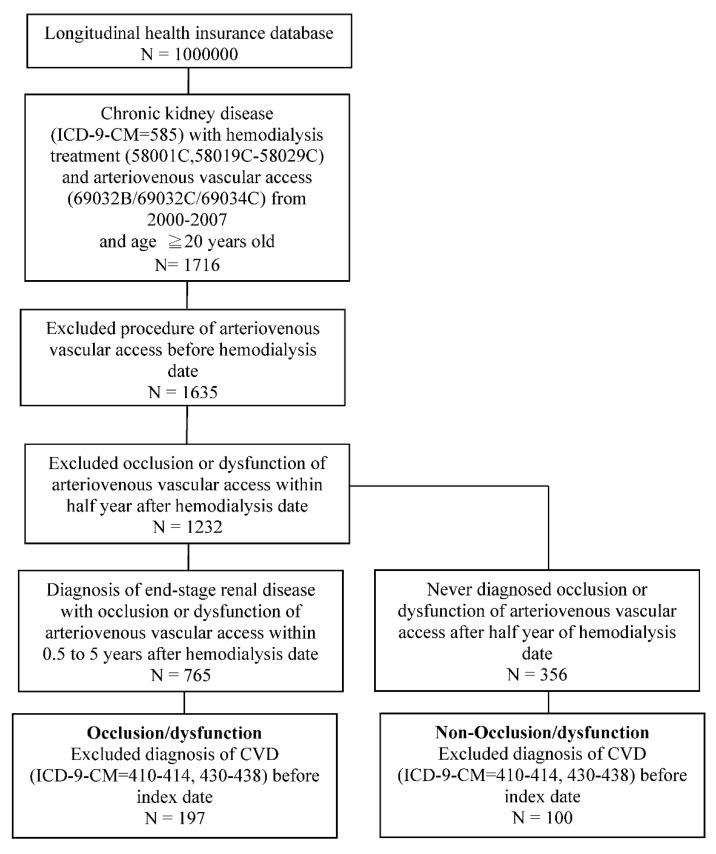
Flowchart describing the selection of VA dysfunction and non-dysfunction. VA group. VA, vascular access.

**Figure 2 jpm-12-00598-f002:**
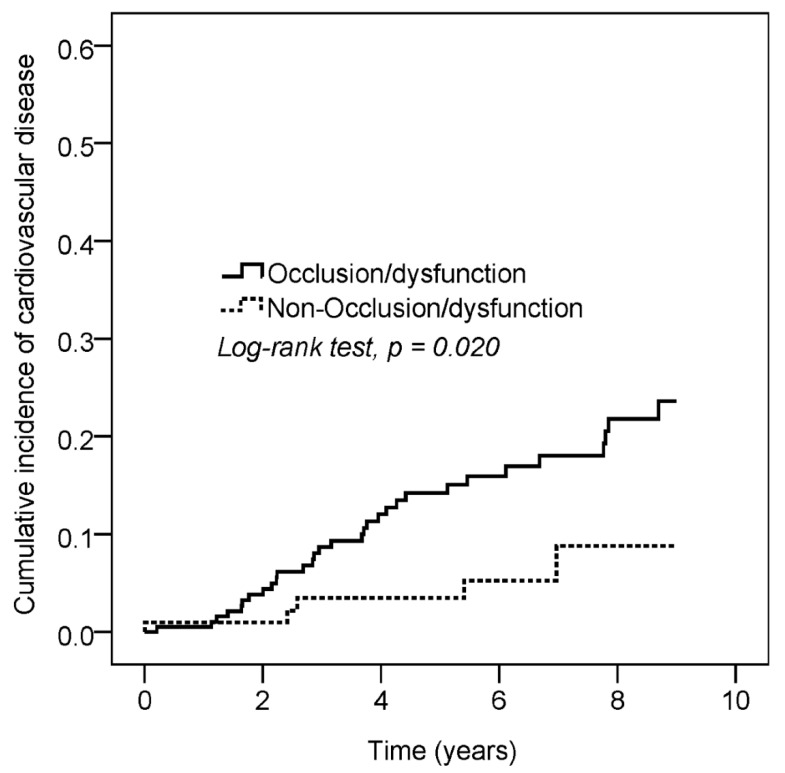
The Kaplan–Meier analysis revealed that the cumulative incidence of cardiovascular disease was higher in the permanent VA dysfunction group than in the comparison group (*p* = 0.020).

**Table 1 jpm-12-00598-t001:** Demographic characteristics of the patients.

	Occlusion/Dysfunction (N = 197)		Non-Occlusion/Dysfunction (N = 100)		*p*-Value
	n	%	n	%	
Age					0.009
20–40	31	15.7	6	6.0	
40–65	113	57.4	53	53.0	
≥65	53	26.9	41	41.0	
Mean ± SD	56.2 ± 15.2		62.3 ± 15.7		0.001
Gender					0.588
Female	107	54.3	51	51.0	
Male	90	45.7	49	49.0	
Hypertension	115	58.4	59	59.0	0.918
Hyperlipidemia	31	15.7	7	7.0	0.033
Diabetes	57	28.9	23	23.0	0.276
Heart failure	33	16.8	6	6.0	0.010
COPD	18	9.1	7	7.0	0.531
Autoimmune disease ^†^	12	6.1	1	1.0	0.067
Asthma ^†^	6	3.0	3	3.0	1
Chronic liver disease	45	22.8	20	20.0	0.576
Hyperparathyroidism	17	8.6	3	3.0	0.067
Warfarin	15	7.6	1	1.0	0.017
Corticosteroids	67	34.0	29	29.0	0.383
Statin	33	16.8	8	8.0	0.039
Aspirin	29	14.7	9	9.0	0.163

^†^ Fisher’s exact test. COPD: chronic obstructive pulmonary disease.

**Table 2 jpm-12-00598-t002:** Cox proportional hazard model.

	No. of Cardiovascular Disease	Observed Person-Years	Incidence Density (per 1000 Person-Years)	Crude HR	95% CI	Adjusted HR ^†^	95% CI
Occlusion/dysfunction							
No	5	508	9.8	1		1	
Yes	31	1062	29.2	2.92	1.13–7.54	3.05	1.14–8.20
Age							
20–40	4	214	18.7	1		1	
40–65	23	863	26.7	1.46	0.50–4.22	1.42	0.44–4.53
≥65	9	493	18.3	1.01	0.31–3.27	1.78	0.49–6.50
Gender							
Female	17	847	20.1	1		1	
Male	19	723	26.3	1.32	0.69–2.54	1.04	0.50–2.13
Hypertension	23	855	26.9	1.51	0.76–3.00	1.79	0.85–3.78
Hyperlipidemia	7	182	38.6	1.87	0.82–4.27	3.33	1.07–10.36
Diabetes	6	401	15.0	0.59	0.25–1.43	0.40	0.15–1.06
Heart failure	6	193	31.1	1.43	0.60–3.44	0.92	0.34–2.48
COPD	3	120	25.0	1.13	0.35–3.68	0.55	0.11–2.65
Autoimmune disease	3	73	41.1	1.78	0.55–5.83	1.84	0.47–7.24
Asthma	2	30	67.8	3.43	0.82–14.33	3.61	0.55–23.69
Chronic liver disease	6	341	17.6	0.72	0.30–1.74	0.61	0.24-1.56
Warfarin	4	90	44.6	2.06	0.73–5.82	0.97	0.29-3.19
Corticosteroids	4	466	8.6	0.30	0.11–0.85	0.24	0.08–0.71
Statin	4	199	20.1	0.86	0.31–2.45	0.36	0.09–1.49
Aspirin	10	169	59.1	3.30	1.58–6.87	2.94	1.24–6.98

COPD: chronic obstructive pulmonary disease. ^†^ Adjusted for age, gender, hypertension, hyperlipidemia, diabetes, heart failure, stroke, COPD, autoimmune disease, asthma, chronic liver disease, warfarin, corticosteroids, statin, and aspirin.

**Table 3 jpm-12-00598-t003:** Subgroup analysis of Cox proportional hazard model.

	Occlusion/Dysfunction		Non-Occlusion/Dysfunction		HR	95% CI
	N	No. of Cardiovascular Disease	N	No. of Cardiovascular Disease		
Age						
20–65	144	25	59	2	4.37	1.03–18.51
≥65	53	6	41	3	1.75	0.44–6.99
*p* for interaction = 0.321						
Gender						
Female	107	14	51	3	2.18	0.62–7.60
Male	90	17	49	2	4.19	0.96–18.22
*p* for interaction = 0.286						
Warfarin						
No	182	27	99	5	2.74	1.05–7.14
Yes	15	4	1	0	NA	NA
Corticosteroids						
No	130	27	71	5	2.60	0.996–6.77
Yes	67	4	29	0	NA	NA
Statin						
No	164	28	92	4	3.54	1.24–10.11
Yes	33	3	8	1	0.89	0.09–8.65
*p* for interaction = 0.285						
Aspirin						
No	168	22	91	4	2.64	0.91–7.71
Yes	29	9	9	1	2.55	0.32–20.13
*p* for interaction = 0.019						

## Data Availability

The present study was a retrospective cohort study using the National Health Insurance Research Database (NHIRD).

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
