# Peer review of "Association of Permanent Vascular Access Dysfunction with Subsequent Risk of Cardiovascular Disease: A Population-Based Cohort Study"

_jpm, 2022, doi:10.3390/jpm12040598_

Round 1

Reviewer 1 Report

I had the chance to review the manuscript titled “Association of Permanent Vascular Access Dysfunction with Subsequent Risk of Cardiovascular Disease: A Population-Based Cohort Study.” The manuscript targets an interesting and rarely studied association of vascular access dysfunction and cardiovascular disease. I have a few comments for the author’s consideration.

Minor comments

  1. I would recommend the Material and method section to be limited to your study design. Under the Study group and outcome, the descriptions of previous studies could be moved to other sections of the manuscript.
  2. The authors discuss prior studies that do not show a role of aspirin, statin and other medications for patency of the AVF and AVG. However, it would be beneficial if they could include their thoughts on the use of these medications for CVD in this population (or the subpopulation with permanent VA dysfunction) given the results of their study.
  3. The authors mention improving surveillance for VA patency. They could also include a discussion of CVD surveillance in patients with permanent VA dysfunction.

Author Response

Responses to reviewers’ comments:

#1. I would recommend the Material and method section to be limited to your study design. Under the Study group and outcome, the descriptions of previous studies could be moved to other sections of the manuscript.

Answer: Thank you for your suggestion. We had moved the paragraph” Many studies have excluded primary failures in analyses of patency, while other studies have reported patency only at specific time points (i.e., six months or one year) [16-18]. A large, randomized controlled trial published by the National Institutes of Health Dialysis Access Consortium in 2008 reported that 60% of AVFs failed to sufficiently mature for successful dialysis four to five months after creation [19]” to Introduction section (Introduction, page 5, line 9-14).

#2. The authors discuss prior studies that do not show a role of aspirin, statin and other medications for patency of the AVF and AVG. However, it would be beneficial if they could include their thoughts on the use of these medications for CVD in this population (or the subpopulation with permanent VA dysfunction) given the results of their study.

Answers: Thank you for your suggestion. We had added the subgroup analysis of medication (Table 3).

Table 3. Subgroup analysis of Cox proportional hazard model

Occlusion/dysfunction

Non-Occlusion/dysfunction

N

No. of  cardiovascular disease

N

No. of  cardiovascular disease

HR

95% CI

Age

20-65

144

25

59

2

4.37

1.03-18.51

≧65

53

6

41

3

1.75

0.44-6.99

p for interaction=0.321

Gender

Female

107

14

51

3

2.18

0.62-7.60

Male

90

17

49

2

4.19

0.96-18.22

p for interaction=0.286

Warfarin

No

182

27

99

5

2.74

1.05-7.14

Yes

15

4

1

0

NA

NA

Corticosteroids

No

130

27

71

5

2.60

0.996-6.77

Yes

67

4

29

0

NA

NA

Statin

No

164

28

92

4

3.54

1.24-10.11

Yes

33

3

8

1

0.89

0.09-8.65

p for interaction=0.285

Aspirin

No

168

22

91

4

2.64

0.91-7.71

Yes

29

9

9

1

2.55

0.32-20.13

p for interaction=0.019

#3. The authors mention improving surveillance for VA patency. They could also include a discussion of CVD surveillance in patients with permanent VA dysfunction.  

Answer: The authors thank for your wonderful commend. It is possible that there is an association between VA dysfunction and CVD. The relationship between VA and CVD exists at different levels, ranging from inflammation promoting atherosclerotic disease to vascular remodelling changes of stenosis formation and left ventricular hypertrophy. The authors suggests that the CVD surveillance in patients with permanent VA dysfunction is important. (Discussion, page 13, line 16- page 14 line 2).

Reviewer 2 Report

The authors touched on an interesting topic in their article "Association of Permanent Vascular Access Dysfunction with Subsequent Risk of Cardiovascular Disease: A PopulationBased Cohort Study". 

They screened a vast population and used a reliable database. 

I have a couple of remarks: 

  • Could you comment on the difference in age between the groups? As you may know, CVD risk is quite strictly connected to the patient's age. 
  • You didn't enclose any data about heart failure, previous CV procedures among your patients (e.g., left ventricular ejection fraction, PCIs before the hemodialysis start). As you showed, you had more HF patients in the occlusion group, which may affect the results. 
  • ICD-9 is quite an old version of the classification. I understand that it was still used in your database, but I found it difficult to decode the numbers to check which CVD diagnoses you meant. Could you please provide more detailed information? It would be easier to understand your study. 

Author Response

Responses to reviewers’ comments:

#1. Could you comment on the difference in age between the groups? As you may know, CVD risk is quite strictly connected to the patient's age.

Answer: Thanks for your recommendation. Increased age was associated with shorter primary patency. In CKD patients on hemodialysis, older age confers a CVD risk that parallels the relationship described in the general population. In the study, the association between VA dysfunction and CVD was valid even after adjustment of age parameters. The present results suggested that patients within 20–65 years of age require increased surveillance of their VA patency, as permanent VA dysfunction with CVD was found to be most significant in this age group. The authors added the description in the discussion (Discussion, page 15, line 4-10 )

#2. You didn't enclose any data about heart failure, previous CV procedures among your patients (e.g., left ventricular ejection fraction, PCIs before the hemodialysis start). As you showed, you had more HF patients in the occlusion group, which may affect the results.

Answers: Thank you for your excellent suggestion. It is sorry that we could not obtain data about heart failure and previous CV procedures in the cohort (e.g., left ventricular ejection fraction, PCIs before the start of hemodialysis) due to the limitation of health insurance database. However, we adjust for potential confounders related to CVD (ie, heart failure, warfarin, statin, and aspirin). The incidence rates of CVD in the permanent VA dysfunction and comparison groups were 29.2 and 9.8 per 1,000 person years, respectively. Compared with the control group, the adjusted hazard ratio for CVD for the permanent VA dysfunction group was 3.05 (95% CI: 1.14–8.20; Table 2). CVD was still significantly associated with a prior history of vascular access dysfunction.

#3. ICD-9 is quite an old version of the classification. I understand that it was still used in your database, but I found it difficult to decode the numbers to check which CVD diagnoses you meant. Could you please provide more detailed information? It would be easier to understand your study.  

Answer: The authors thank for your wonderful suggestion. Among the 36 patients with CVD events, 26 patients had coronary artery disease (ICD-9-CM=410-414) and 10 patients had stroke (ICD-9-CM=430-437). The authors added the description in the discussion (Results, page 10, line 13-14)

Round 2

Reviewer 2 Report

Thank you for your answers. I have no more comments.